# The resource team: A case study of a solitary confinement reform in Oregon

David H. Cloud[1]*, Craig Haney[2], Dallas Augustine[1], Cyrus Ahalt[1], Brie Williams[1]*

**1** Center for Vulnerable Populations, School of Medicine, University of California San Francisco, San Francisco, CA, United States of America, **2** Department of Psychology, University of California, Santa Cruz, Santa Cruz, CA, United States of America

* david.cloud@ucsf.edu (DHC); brie.williams@ucsf.edu (BW)

**Data Availability Statement:** "This study utilized both administrative and qualitative data. Administrative data cannot be shared publicly because the administrative data the authors analyze in this paper is drawn from a confidential

## Abstract

The continued use of solitary confinement has sparked international public health and human rights criticisms and concerns. This carceral practice has been linked repeatedly to a range of serious psychological harms among incarcerated persons. Vulnerabilities to harm are especially dire for persons with preexisting serious mental illness ("SMI"), a group that is overrepresented in solitary confinement units. Although there have been numerous calls for the practice to be significantly reformed, curtailed, and ended altogether, few strategies exist to minimize its use for people with SMI and histories of violence against themselves or others. This case study describes the "Oregon Resource Team" (ORT), a pilot project adapted from a Norwegian officer-led, interdisciplinary team-based approach to reduce isolation and improve outcomes for incarcerated persons with SMI and histories of trauma, self-injury, and violence against others. We describe the ORT's innovative approach, the characteristics and experiences of incarcerated people who participated in it, its reported impact on the behavior, health, and well-being of incarcerated persons and correctional staff, and ways to optimize its effectiveness and expand its use.

## Introduction

Solitary confinement is a grave public health and human rights concern. In the United States, approximately 4.5% of people in state prisons are housed in solitary confinement—between 55,000–62,000 people—where they experience social isolation, enforced idleness, and material deprivation [1]. They remain locked in their cells for upwards of 22 hours per day, confined to spaces typically no larger than a wheelchair-accessible restroom, where they eat, sleep, and use the toilet. Their access to outdoor recreation is often limited to an hour a day, and commonly takes place in small, caged-in areas. People are usually placed in physical restraints (e.g., handcuffs, ankle shackles, waist chains) whenever they are escorted out of their cell. Although the essence of solitary confinement is the deprivation of meaningful social contact, there are typically additional, severe restrictions placed on personal property, reading materials, and access to education and programs [2–4]. People with serious mental illness ("SMI") are disproportionately exposed to and harmed by this penological practice [5, 6]. Indeed, preventing

data file, shared with the research team for the limited purpose of evaluating the Resource Team pilot in the Oregon department of corrections. If any researchers wish to obtain similar data from the Oregon department of corrections, the authors of this paper would be willing to consult with those researchers about the request and the process for obtaining the data. In theory, the administrative data file used in this study could be accessed again by future researchers. External researchers can also access the Oregon Department of Corrections Research Committee form on the agency's website (https://www.oregon.gov/doc/Forms/cd-1838-research-application.pdf) to learn about the legal requirements and application process. The authors are also precluded from publicly sharing qualitative data. These transcripts contain sensitive information for correctional staff and incarcerated people, and the protocol was designed to protect the privacy of this medically and socially vulnerable group. Providing additional information beyond anonymized quotations reported in the findings (such as full transcripts or data sets) would enable identification of the participants, especially when combined with administrative data. This would breach participants' confidentiality under terms of informed consent and protocol of the University of California Institutional Review Board, and data agreements with the Oregon Department of Corrections Research Committee."

**Funding:** This research was supported by the Jacob and Valeria Langeloth Foundation, The Charles and Lynn Schusterman Family Philanthropies, Arnold Ventures, and other private donors. The funders had no role in study design, data collection and analysis, decision to publish, or preparation of the manuscript.

**Competing interests:** Authors DC, DA, CA, and BW are members of the Amend team at UCSF. CA, BW, and CH co-facilitated the Norwegian exchange program and provided the technical assistance to develop the Resource Team intervention.

exposures to solitary confinement for people with SMI is a central focus of advocacy, litigation, legislation, and system-initiated reforms. Yet, even when efforts to adopt new laws and policies designed to restrict the use of solitary confinement are successful, they often face a long and challenging implementation process before bringing about transformative change. A complex implementation process must follow to translate new policies into practice. Despite widespread calls to significantly restrict, reform, or end the use of solitary confinement [7, 8], correctional agencies have struggled to heed them, even in jurisdictions with correctional officials more receptive to change [9–11]. A variety of structural, legal, organizational, and cultural factors may influence whether and how solitary confinement reform efforts succeed in achieving their objectives.

Because only a few attempts to significantly reform the practice have been empirically assessed [9, 10, 12], additional studies are needed to build upon an important, emerging body of scholarship focused on assessing whether and how reform efforts achieve their intended objectives. For example, one case-study unpacked the policy changes that North Dakota correctional leaders initiated, which achieved significant reductions in the use of solitary confinement, and were credited with decreasing violence and other benefits for incarcerated persons and staff alike [12]. Schlanger (2020) utilized an "incrementalist versus maximalist" framework to identify the policy levers and institutional factors that may facilitate or impede reforms from achieving their stated goals [11]. Similarly, Augustine et al. (2021) reported on successes and limitations of initiatives to improve conditions within solitary confinement units in Washington prisons [9]. The present study builds on this emerging body of scholarship focused on assessing whether and how reform efforts achieve their intended objectives.

Research has repeatedly linked solitary confinement to a range of serious psychological and physical harms, including anxiety, depression, hypertension, psychosis, self-harm, and suicidality [4, 13–15], especially among persons with SMI [6, 14]. Since most people exposed to solitary confinement are released from prison [16], the practice can adversely affect community health and public safety. Indeed, exposure to solitary confinement has been associated with post-release morbidity and mortality due to suicide, homicide, overdose [17–19], and re-incarceration [20].

Correctional staff are not immune to the harmful effects of solitary confinement. Working in these environments represents an occupational hazard that can imperil the health, safety, and well-being of correctional officers [21] who, as a group, already experience a lower than average life-expectancy, and elevated rates of poor health outcomes, including alcoholism, depression, and suicide [22–25].

In the United States, correctional staff typically receive limited if any training about how best to respond to the manifestations of psychological deterioration they witness occurring among incarcerated persons (e.g., smearing feces, ingesting objects, self-injury, violent outbursts). Not surprisingly, most officers react to these situations in the ways in which they have been trained—through the use of physical force (e.g., cell extractions, chemical spray, tasers) and the imposition of additional punishment (e.g., taking away property, phone calls, and/or yard time, extending time in solitary) [26, 27]. Not only are these reactive and aggressive responses often counterproductive (engendering more suffering and decompensation rather than less), but they also carry the risk of physical and psychological injury for both incarcerated persons [4, 13, 26] and correctional staff [28, 29]. Moreover, they increase the likelihood of staff experiencing vicarious trauma that results from witnessing intense human suffering and interpersonal conflicts, and the moral injuries incurred from whatever role they played in initiating, maintaining, or exacerbating these events [21, 28].

Concerns over the use of solitary confinement are not limited to the United States. Even nations committed to operating more humane carceral regimes have been criticized on

human rights grounds [8, 30]. Thus, the Norwegian Correctional Service, internationally recognized for its humane prison system overall, was subjected to critical scrutiny for its solitary confinement practices, especially for people with SMI and histories of violence against others [31]. The Norwegian Parliamentary Ombudsman issued a report in 2019 expressing concerns about excessive uses of solitary confinement and called for policy changes to ensure that incarcerated persons who are separated from the general prison population were provided with enhanced structured activities and daily meaningful human contact [32]. In direct response to these criticisms, the Norwegian correctional service developed a novel approach as one way to minimize prolonged isolation and its harmful effects: the Resource Team (or "RT.").

This case study describes principles and practices of the Norwegian RT and their adaptation to one maximum-security U.S. prison, the Oregon State Penitentiary. Our discussion of the Oregon Resource Team ("ORT") pilot program explains and assesses its operation, including: (1) the characteristics and experiences of the incarcerated persons who participated in it; (2) the nature and scale of activities and services that were facilitated by ORT staff as part of the program; (3) the impacts of the ORT on the behavior, health, and well-being of the prisoner and correctional staff participants.

## The Norwegian resource team

The Norwegian RT was developed by the Norwegian Correctional Service to improve rehabilitative outcomes for the highest risk, highest need persons incarcerated in the nation's prisons. It is an interdisciplinary team-based approach to increasing out-of-cell time and providing enhanced activities and other services to persons whom prison officials regard as posing the highest immediate safety risk to themselves or others. The operation of the RT is grounded in three key principles: "dynamic security" (nurturing positive interpersonal relations between staff and incarcerated persons), "progression" (continuously working towards transitioning incarcerated persons to the least restrictive environment possible), and "normalization" (creating living conditions in correctional settings that resemble the larger outside community as closely as possible) [12]. The RT provides people who remain separated from mainline prison settings with frequent opportunities to engage in meaningful social interaction, physical exercise, and clinical services with the aim of eliminating or minimizing correctional systems' reliance on static-security measures, such as restraints and uses-of-force.

Norwegian RT officers receive advanced training on the effects of isolation, the nature of mental illness and trauma, conflict resolution and de-escalation, and ways to motivate incarcerated persons to engage in programming and mental healthcare. These officers lead an interdisciplinary team (including medical and mental healthcare staff) to develop individualized program plans designed to eliminate (or drastically reduce) the use of solitary confinement for each person. The RT goals are to (1) maximize out-of-cell time in which persons are engaged in meaningful social activity, programming, and treatment; (2) support each person to live in the lowest level of security possible; and (3) enhance the safety of incarcerated persons, correctional staff, and society by focusing on improving the health and dignity of each person.

## The Oregon Resource Team

Solitary confinement reforms in the Oregon Department of Corrections ("ODOC") began much as Norway's did—in response to mounting allegations of inhumane treatment of people with serious mental illnesses who were held in an extremely harsh solitary confinement-type housing unit. The Behavioral Health Unit ("BHU") at the Oregon State Penitentiary, is a separate unit located on the prison grounds, designed to house persons whom prison officials have deemed to be among the most vulnerable and disruptive in their custody. This designation is

based largely on serious and lengthy mental health histories and past instances of violence against themselves or others. Many had frequent and lengthy prior stays in solitary confinement that often resulted in psychological deterioration. Some had reached the point where they refused to leave their cells and declined to participate in any of the already limited programming available to them.

A 2015 investigation of the BHU conducted by Disability Rights Oregon ("DRO") reported that people were being kept in "tiny, stifling cells" for up to 23 hours per day, receiving few psychiatric services, and that the unit was pervaded by a "culture that promotes unnecessary violence and retaliation by correctional staff" (Greenberg & Radcliffe 2015:1). In response, the ODOC signed a memorandum of understanding with DRO to increase out-of-cell time, provide more confidential therapeutic treatment, and reduce uses of force by officers. However, by 2018, despite some improvements, many goals had not been achieved. Although out-of-cell time increased between 2015 and 2018, it remained below the overall 20 hour weekly goal set by the parties. DRO also had encouraged ODOC to develop a "therapeutic intervention" focused on reducing "refusals to leave cell" [33], but the ODOC had failed to achieve this goal.

In 2017, amid ongoing concerns over the operation of the BHU, ODOC officials participated in a public health-focused prison culture transformation program in Norway led by Amend, a group of academics based in the School of Medicine at the University of California San Francisco. Drawing on principles of public health, medical ethics, occupational health, and human rights, Amend seeks to address the debilitating health effects of incarceration on people who live and work in prisons [12, 30]. To further these goals, Amend provides multiyear, immersive programs in Norway and elsewhere that include correctional staff training, policy review and revision, mentored culture change initiatives, technical assistance, and evaluation support.

In 2018, ODOC staff who had participated in Amend's immersion program in Norway began planning to launch a Resource Team at the Oregon State Penitentiary. Like the Norwegian counterpart on which it was modeled, the primary aims of the Oregon Resource Team (or ORT) were to reduce time spent in isolation by persons who were both SMI and had histories of chronic violent behavior in prison. The ORT sought to create a range of carefully structured, individualized opportunities to participate in meaningful social activities led by trained correctional officers.

The ORT began in the first quarter of 2019, initiated by a small group of OSP correctional officers who were trained by Norwegian officers, through classroom-based instruction and "job shadowing" opportunities in Norway and Oregon. The training focused on fostering positive social interactions with otherwise isolated persons, conducting immediate and ongoing individualized risk assessments, preventing violence by de-escalating conflict, planning safe but meaningful social activities, and motivating participants to participate in available programming and engage constructively with healthcare and social services staff. In a departure from the Norwegian model, the ORT eventually developed a "peer mentoring" component that employed incarcerated persons who had experienced solitary confinement to assist correctional officers in planning and participating in activities, building trust between staff and participants, and providing program participants with social and emotional support.

The program began on a very small scale by focusing on just two persons who had been chronically isolated in the BHU. Based on its initial, albeit limited success, the ORT worked in tandem with Amend and their Norwegian colleagues to create new policies and procedures to facilitate program expansion. Starting in 2019, a team of ORT officers identified additional incarcerated persons in the BHU as potential participants. Selection was based on several factors, such as whether the person had experienced frequent and/or prolonged solitary confinement, had engaged in interpersonal violence with peers and/or staff, had instances of self-

injury or suicidality, and/or had chronically declined or refused to leave their cell for programming, showering, or recreation. For example, people who were housed in other solitary confinement units in the ODOC, such as the "Intensive Management Units" at Two Rivers Correctional Facility and Snake River Correctional Institution, and who had recently been in situations involving acute psychiatric decompensation, self-injury, and/or multiple assaults against staff or other incarcerated persons that often escalated into uses of force by officers (e.g., cell extractions, chemical spray), before being transferred to the BHU were among those considered for inclusion. Although the pandemic limited the team's ability to recruit new participants and conduct additional activities, by 2021, five full-time officers were funded to work with mental health and medical staff on an interdisciplinary ORT.

This case study describes and evaluates the ORT using institutional data and the self-reported experiences of prison staff and incarcerated persons who participated in the ORT during the first nine months that it was fully staffed (from January 2021 through September 30, 2021).

## Methods

### Methodological approach

We adopted a mixed-method/multiple source case-study design to describe the scope and components of the ORT and the experiences of incarcerated persons and staff members who were involved in the project [34]. Our analysis is based on descriptions of the changes brought about by the ORT according to staff and incarcerated persons, including the nature and effects of solitary confinement before the ORT began, and observations of changes in the interactions and atmosphere in the housing units once the ORT began. We triangulated these qualitative assessments with institutional-level data describing staff uses-of-force and individual-level data describing self-injurious behavior or violence against others before and after the ORT reforms were implemented. The University of California San Francisco Internal Review Board and Oregon Department of Corrections Research Committee approved the activities for this study described below.

### Study setting and participants

As noted, the ORT project was piloted in the special housing units at OSP, the Oregon prison system's main penitentiary, with a primary focus on the Behavioral Health Unit (BHU). The BHU is one of several special housing units in operation at OSP, all of which are intended for persons who are deemed "unable to adjust satisfactorily to the general population because of a serious mental illness." The BHU (49 cells) is designated for "intensive behavioral management and skills training unit for inmates with serious mental illness that have committed violent acts or disruptive behavior," all of whom are considered "unable to adjust satisfactorily to the general population because of a serious mental illness." Another special housing unit, the Mental Health Infirmary (49 beds) is an acute "crisis response unit that provides psychiatric care and a therapeutic environment for inmates [sic] that require intensive assessment, care, and stabilization." A third such unit, the Intermediate Care Housing (45 beds) is a "step down" unit for people transitioning out of the infirmary or BHU after stabilizing. The Day-treatment Unit (40 cells) permits residents to have more unstructured out-of-cell time, access to a dayroom, and is most similar to conditions in the general population. Finally, the Disciplinary Segregation Unit (65 beds) is punitive housing for people charged and convicted of rule violations, OR. Admin. R. 291-048-0210).

The BHU was the main focus of the ORT, although several ORT participants were drawn from other OSP special housing units. It was selected for this intervention because its residents

are considered by OSP staff to be among the most challenging to manage in the Oregon prison system and likely to be retained in long-term isolation. In many instances, their SMI diagnoses had contributed to frequent acts of self-injury and violent behavior. Many residents had been retained in solitary confinement-type conditions for years and had ceased participating in activities or utilizing out-of-cell time (including refusing showers and yard time). A number of them had manifested the extreme negative effects of solitary confinement by engaging in clinically dysfunctional and disruptive behavior (e.g., smearing feces on walls, flooding their cells with toilet or sink water, starting fires). All 14 ORT staff and all 44 incarcerated people who worked with the ORT during the evaluation period (January 1 through September 30, 2021) were eligible for qualitative interviews.

## Data collection and measures

Our data collection included three components:

**Oregon Resource Team weekly logs.** The weekly ORT activity logs document the nature and scope of staff-led activities. We used the logs to determine the weekly frequency with which BHU residents participated in ORT activities during the study period, the exact nature of the activities, and any negative events (e.g., assaults, conflicts) that occurred during these activities.

**Interviews with ORT participants and staff.** We conducted semi-structured interviews with subsamples of the incarcerated persons who participated in the program (interviewed in July 2021) and the staff who devised and implemented it (interviewed in May 2021). Due to COVID-19 related travel restrictions, we were only able to visit the prison to recruit participants for qualitative interviews for two days in July 2021. Of the 44 persons who participated in the ORT during the study period, 38 had begun participating by the time of our July 16, 2021, interviews. Of the 38, 17 had been moved out of the housing units or were not available at the time of our interviews. In addition, as a measure of the degree of vulnerability and impairment of the population of persons who participated in the ORT, four were deemed by healthcare professionals to be too impaired to consent for in-person interviews. Of the remaining 17 ORT participants, one person declined an interview, resulting in a total of 16 ORT participant interviews.

Our semi-structured interview format focused on participants' experiences before and during incarceration, the amount of time spent in solitary confinement, and their experiences with the ORT. Researchers used an interview guide with a series of prompts intended to elicit participants' lived experiences with solitary confinement (e.g. frequencies of admission, reasons for placement in solitary confinement, lengths of stay, access to programming, recreation, treatment etc.) and how those experiences affected their overall well-being. Participants were also asked about their initial and ongoing interactions with the Oregon Resource Team members, and whether and how the program had affected them personally, as well as their experience of the climate and culture in the special housing units. They were also asked to discuss any positive aspects of the program and give feedback on ways to enhance it. All recruitment, consenting, and interviews were conducted in a private office, where no correctional staff were present. For interviews with incarcerated persons, we obtained verbal consent to further protect confidentiality, which was documented by the interviewer. All participants were unrestrained during the interviews, which lasted 60–90 minutes and were audio recorded.

We also conducted interviews with 14 correctional staff members about the ORT. We used a semi-structured interview format to assess staff perceptions of well-being, occupational safety, and overall job satisfaction. Interviews took place in a private setting, averaged 30 to 45

minutes, and were audio recorded. All 14 staff members who were present during our visit provided written consent to be interviewed.

**Administrative data.** We supplemented our interviews with administrative data. First, we used administrative data to further determine the make-up of the 44 persons in our overall participant sample (i.e., demographics, housing assignments, disciplinary records). We also examined "Unusual Incident Reports" that described self-injuries, suicide attempts, medical emergencies, and uses of force (taser, chemical spray, cell extractions). We used housing files to calculate each participants' length of imprisonment and the frequency and duration of time spent in solitary confinement in any ODOC facility, and disciplinary records to identify number of convictions for assaults on staff and other incarcerated people before and after engagement with the ORT. We defined "solitary confinement" as placement in any of several different ODOC facilities where persons are placed in isolation. In addition to the aforementioned BHU at the Oregon State Penitentiary, incarcerated persons in Oregon could be housed in an Intensive Management Unit ("IMU"), which is located at the Snake River Correctional Institution, or in any Disciplinary Segregation Unit ("DSU"), located in most ODOC facilities and used to house persons in response to specific rule violations.

One of the explicit goals of the ORT was to reduce the use of physical force against BHU residents by employing de-escalation tactics and relationship building in response to problematic behavior. Therefore, we used institutional data to calculate staff uses of force in the special housing units before and after implementation of the ORT initiative, plotting unit-level trajectories of use-of-force incidents from January 1, 2016 (three years prior to the launch of the ORT initiative) through that calendar year (December 31, 2021). We chose to examine institutional records for three-years prior to the pilot launch to allow for a more contextualized assessment of ORT participants' involvement in these incidents over longer periods of time. We received a consent waiver for accessing administrative data.

## Data analysis

We plotted quarterly unit-level changes in staff uses of force across OSP's special housing units before (from January 1, 2016 to December 31, 2018) and after (from January 1, 2019 to December 31, 2021) the launch of the ORT. The quarterly mean number of use-of-force incidents was then calculated for each observation period and a paired-t test was conducted to assess the statistical significance of observed changes.

We then compared recorded disciplinary infractions (including self-harm and/or violence against staff or residents) among ORT participants prior to and since engaging with the program. Because staff reported that people with fewer than three ORT contacts either were released or transferred too fast for sustained engagement in the ORT, this analysis focused only on those persons who had at least three contacts with the ORT (n = 31, 70.5% of all ORT participants). We assessed whether people enrolled in ORT incurred a reduced number of disciplinary infractions by calculating the mean rate of disciplinary infractions and assaults (combining assaults on staff and assaults on incarcerated persons into a single indicator due to the small sample size) per 100 incarceration days prior to and after each person's first engagement with the ORT. We compared the mean rates before and after ORT engagement using Wilcoxon Signed-Rank Tests.

We also conducted a thematic analysis of our interviews with ORT participants and staff to determine each group's perceptions of the program's effects on their health and well-being, and to contextualize the administrative data analyses. Our thematic analysis included an iterative process of coding transcripts for emergent themes, and structured deliberations among

co-authors to interpret findings in relation to other components of analysis [35, 36]. All analyses were conducted using NVivo (Version 12) and STATA (Version 17).

## Results

### Population served by the Oregon Resource Team

Of the 133 people housed in any mental health "special housing unit" at OSP between January and September of 2021, 44 (33.1% of total) engaged at least once with the Oregon Resource Team. Although the ORT did not set firm eligibility criteria, its mission was to work with incarcerated persons with one or more of the following attributes: extensive histories of solitary confinement due to disciplinary sanctions; tendencies to self-isolate and refuse recreation, showering, programming, and other types of out-of-cell opportunities; serious mental health diagnoses and/or history of trauma before and during incarceration; repeated self-injurious behavior (non-suicidal self-harm and/or suicide attempts); and frequent or recent involvement in interpersonal violence against staff and other incarcerated persons.

As Table 1 indicates, the mean age of the 44 ORT participants was 36.5 years (range 20s to 60s), their average prison sentence was 10.8 years (SD = 2.7 years). The calculation of average prison sentences excluded the two people with life sentences. Most participants (70.5%) had been incarcerated for less than 5 years, although a small minority (6.8%) had been in prison for 15 or more years. As Table 1 also illustrates, ORT participants demographically resembled the larger group of persons who were housed in the OSP special housing units, with white participants accounting for 79.6% of ORT participants (vs. 74.2% of those eligible, p = 0.42). The only statistically significant difference between the groups occurred with respect to Latino participants, who were significantly underrepresented in the ORT (4.6%) as compared to in special housing overall (7.5%).

As we noted, the ORT participant group was comprised of persons with SMI, and extensive disciplinary histories and amounts of time spent in solitary confinement. According to administrative records, all ORT participants held a diagnosis of at least one SMI. On average, participants had experienced 9.7 admissions to solitary confinement units in ODOC, for a cumulative mean of 625.5 days (range 8–4,786 days) across all admissions. Nearly half (47.2%, n = 17) had spent more time in solitary confinement units than in the general prison population. Nearly three-quarters (73.3%) had been sanctioned, at least once, to a Disciplinary Segregation Unit for violating prison rules during their current incarceration (average 7.8 admissions). Nearly all of the participants (86.7%) had received a misconduct for an assault on another incarcerated person (total of 267 incidents).

More than a quarter (27.3%) had been housed in the other highly restrictive solitary confinement unit in the state system—the Intensive Management Unit (IMU) at Snake River Correctional Institution. The Snake River IMU is among the most restrictive, long-term solitary confinement in the Oregon prison system. The Oregon Department of Corrections designates the IMU for incarcerated people "who are under investigation for or who have been charged with the in-custody murder or assault of another inmate or staff, to special security housing and programs in a designated IMU or IMU status cells separate from general population housing in Department of Corrections facilities to provide the maximum level of inmate security, control, and supervision as provided in these rules." (ODOC 291-055-0005).

Self-injury and suicide attempts were also prevalent among participants. According to administrative records, nearly one-quarter (22.7%) of ORT participants had engaged in at least one act of self-injury that was serious enough to require medical attention during their imprisonment. Over one quarter (26.7%) of the 44 ORT participants had been subjected to at least

**Table 1. Demographics and institutional histories of incarcerated persons contacted by the Oregon Resource Team January—March 2021.**

| Demographics | Total sample No.(%) or Mean ± SD |
|---|---|
| Age | 36.5 ± 9.2 |
| Black/African American | 5 (11.4%) |
| Latino | 2 (4.6%) |
| Native American | 2 (4.6%) |
| Non-Hispanic White | 35 (79.6%) |
| Asian | 0(0%) |
| **Years imprisoned** | |
| Less than 1 year | 6 (13.6%) |
| 1–5 years | 25 (56.8%) |
| 6–10 years | 10 (22.7%) |
| 15 or more years | 3 (6.8%) |
| **Time in Solitary Confinement*** | |
| < 30 days | 15 (34.9%) |
| < 6 months | 8 (18.6%) |
| 6 months-1year | 2 (4.7%) |
| 1–3 years | 10 (23.3%) |
| > 3years | 8 (18.6%) |
| **Admissions to Solitary Confinement** | |
| No admissions | 8 (18.2%) |
| 1–5 admissions | 16 (36.4%) |
| 5–10 admissions | 9 (20.5%) |
| 10–15 admissions | 4 (9.1%) |
| More than 15 admissions | 7 (15.9%) |
| **Assaults on another Incarcerated Person** | |
| None recorded | 6 (13.6%) |
| 1–5 assaults | 19 (43.2%) |
| 6–10 assaults | 10 (22.7%) |
| 10–15 assaults | 6 (13.6%) |
| More than 15 assaults | 3 (6.8%) |
| **Assaults on Staff** | |
| None recorded | 25 (56.8%) |
| 1–5 assaults | 13 (29.6%) |
| 10–15 assaults | 5 (11.4%) |
| More than 15 assaults | 1 (2.3%) |
| **Exposure to Other Physical Violence** | |
| Any self-injurious behavior | 10 (22.7%) |
| Subjected to Use of Force | 10 (22.7%) |

*Solitary Confinement refers to placements in Disciplinary Segregation Unit; Intensive Management Unit; and/or Administrative Segregation Units.

one documented use-of-force incident during their current incarceration, for a total of 53 incidents.

## Enriching the BHU environment

The ORT was focused on engaging its chronically isolated, seriously mentally ill participants in productive out-of-cell time, including proactively encouraging them to engage in meaningful

interpersonal interactions and activities. Between January 1 and September 30, 2021, the weekly ORT logs reflected a total of 290 recorded, individualized interactions between ORT staff and the 44 program participants. Weekly engagements averaged 6.6 per person (range 1 to 25 interactions). Staff reported facilitating between 3 and 5 activities each day, which included activities conducted on a one-to-one, small group, or congregate (e.g. art class, movies) basis.

The ORT activities were guided by an assessment of each person's needs, goals, and interests, with the goal of promoting meaningful social interactions, physical exercise, and emotional stability. These activities ranged from staff playing sports with participants (e.g., basketball, cornhole tournaments, cross-fit competitions), playing board games, attending movie screenings, watching sports events, and sharing meals. The ORT also arranged trips out of the BHU to visit congregate settings within the larger prison (e.g., day rooms, main prison yard, library, cafeteria) to acclimate them to these social situations and foster confidence in their social skills. ORT staff created group art classes and encouraged participants to write, record, and/or perform music.

## Reductions in staff use-of-force incidents

In addition to promoting positive interactions and activities, the ORT sought to decrease the number of negative interactions between residents and staff, especially to reduce staff uses of force. As in most prison systems, ODOC staff members assigned to special housing units had been trained to respond to troublesome behaviors (e.g., feces smearing, refusing medication and showers, self-injury) by employing cell extractions, chemical agents, tasers, and/or mechanical restraints. Yet these can be and often are traumatic, both to the persons who directly experience them as well as those who witness them. Indeed, the incarcerated persons we interviewed described these experiences as "dehumanizing," and acknowledged that they often triggered downward spirals of feeling re-traumatized, experiencing paranoia and depression, and counter-productively leading to increased self-isolation, self-injury, and further confrontations with staff.

Combined data from the BHU and OSP's other special housing units suggest that the Oregon Resource Team had a broad impact across these units. According to administrative data, as Table 2 documents, compared to the previous three years, in the three years since the ORT was launched in 2019, the mean of uses-of-force significantly decreased (from 10.7 to 2.8, $p < .01$) across all special housing units at OSP. As would be expected, most of the overall reductions are accounted for by the dramatic decreases in the quarterly occurrence of use-of-force incidents that occurred in the BHU (from an average of 5.6 to 0.8 incidents per quarterly period, $p < .01$), representing an 85.7% decrease. The dramatic nature of the overall reduction in uses-of-force is visually illustrated in Fig 1.

Fig 1 displays quarterly incidents in uses-of-force for all special housing units at the Oregon State Penitentiary from January 2016 through December of 2021. It includes data for all residents in these units to depict unit-level changes, which may be in part due to the implementation of the Resource Team.

## Reductions in disciplinary infractions for violence

Another goal of the ORT was to positively impact the behavior of the incarcerated participants. To evaluate one important behavioral change, we calculated mean rates of disciplinary infractions for assaults among ORT participants (combining assaults on staff or other residents) since 2016 and compared mean rates of assaults per 100 incarceration days prior to and after each person's engagement with the ORT. As Table 3 documents, we found that among the subgroup of ORT participants who had at least 3 interactions with the ORT, there was a 55.7%

**Table 2. Mean quarterly use-of-force incidents pre vs. post implementation of the Oregon Resource Team (ORT) (January 1, 2016- December 31, 2021).**

| | Pre-ORT | Post-RT | | |
| --- | --- | --- | --- | --- |
| | Mean (SD) Use-of-Force Incidents | Mean (SD) Use-of-Force Incidents | % Change | p-value |
| Units Combined | 10.7 (5.6) | 2.8 (4.7) | -73.8% | <0.01* |
| Behavioral Health Unit (BHU) | 5.6 (3.3) | 0.8 (1.14) | -85.7% | <0.01* |
| Mental Health Infirmary (MHI) | 2.9 (2.2) | 1.25 (2.3) | -56.9% | 0.10 |
| Disciplinary Segregation Unit (DSU) | 1.4 (1.6) | 0.5 (1.2) | -64.3% | 0.12 |
| Intermediate Care Housing (ICU) | 0.8 (0.9) | 0.3 (0.9) | -62.5% | 0.18 |

Table 2 compares the mean use-of-force incidents on a quarterly basis for each type of unit in the special housing wing of the Oregon State Penitentiary, before (from January 1, 2016-December 31, 2018) and after (January 1, 2019-December 31, 2021) the implementation of the Oregon Resource Team intervention. The Behavioral Health Unit (BHU, 49 cells) is designated for people "intensive behavioral management and skills training unit for inmates [sic] with serious mental illness that have committed violent acts or disruptive behavior." The Mental Health Infirmary (MHI, 49 beds) is an acute "crisis response unit that provides psychiatric care and a therapeutic environment for inmates [sic] that require intensive assessment, care, and stabilization." The Disciplinary Segregation Unit (DSU, 65 beds), a form of punitive solitary confinement for people charged and convicted of rule violations, is also located on the same wing as the mental health units but is mostly comprised of people residing in the prison's general population The Intermediate Care Housing (ICH, 45 beds) is a "step down" unit for people transitioning out of the infirmary or BHU after stabilizing. (OR. Admin. R. 291-048-0210).

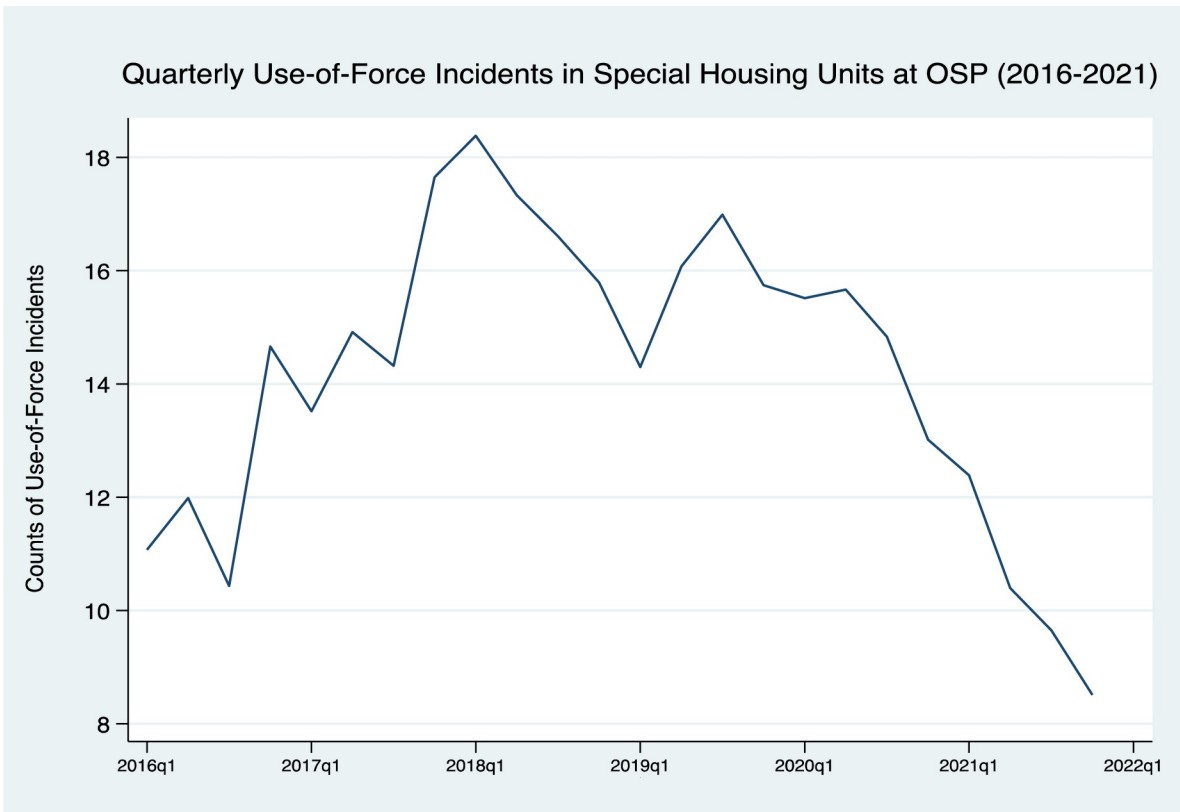

**Fig 1. Decline in quarterly use of force following introduction of Oregon Resource Team.** Fig 1 displays quarterly incidents in uses-of-force for all special housing units at the Oregon State Penitentiary from January 2016 through December of 2021. It includes data for all residents in these units to depict unit-level changes, which may be in part due to the implementation of the Resource Team.

**Table 3. Rates of total disciplinary infractions, assaults (per 100 incarceration days): Pre vs post resource team engagement.**

| | Pre-RT Engagement | Post-RT Engagement | % change | p-value |
|---|---|---|---|---|
| Mean Rate of Disciplinary Infractions (per 100 incarceration days) | 1.31 | 0.58 | -55.7% | p = 0.002* |
| Mean Rate of Assaults (per 100 incarceration days) | 0.46 | 0.12 | -73.9% | p = 0.001* |

Disciplinary infractions include at major and minor incidents where a person was convicted of violating any prison rule. Assaults included a combined endpoint of assaults against staff and/or other incarcerated people. Counts of disciplinary infractions and assaults were converted to rates per 100 incarceration days for individuals enrolled in ORT.

reduction in mean rate of disciplinary infractions in general and a 73.9% decrease in the mean rate of assaults in particular. Both reductions were statistically significant ($p < 0.01$).

## Impact of the Oregon Resource Team on the health and well-being of residents

**Promoting health among incarcerated people.** The incarcerated persons who participated in the ORT consistently reported that it had a positive impact on their health and well-being. Some acknowledged that engaging in simple activities such as dominoes, basketball, and art projects helped them to overcome the anxiety, depression, and psychotic symptoms they experienced in solitary confinement. One participant explained:

> When I'm locked in there [BHU], I sometimes feel really antsy. . .desperately depressed, like there's a huge hole and it hurts. And I need to get out, get away. So the Resource Team, they come to my rescue at times and do stuff with me.

Others reported developing trust and social bonds with ORT staff members, which they felt provided emotional support. Several said that the professional relationships they forged with ORT staff members helped them to cope with sensitive personal issues, such as grieving the death of a loved one, processing childhood trauma, or preventing or reducing self-injurious behavior. For example, one participant said that ORT staff had gained his trust by showing him empathy and compassion, which he found helpful and healing:

> Sometimes a person just needs to know that they are cared for, and that they have a place, even if that life has been temporarily shattered. So, while the Resource Team can't heal all of the trauma, they can help slowly and methodically to get past some of the pain.

Because solitary confinement not only deprives people of social contact but also typically precludes their contact with the natural environment, the increased opportunities to reconnect with nature that the ORT provided were especially poignant for many participants. They described the importance of being given access to greenery, natural sunlight, fresh air, and companionship in a "Japanese Healing Tea Garden" that was created at the OSP in 2019, following ODOC's introduction to the Norwegian principle of normalization in prisons (i.e., creating correctional conditions that resemble the outside world as closely as possible) and progression (continuously moving incarcerated persons to less restrictive environments) [12, 30, 37]. One person who had spent twenty years in isolation before working with the ORT described going to the garden as the "first time I walked on grass in twenty years." As he stated:

> In an environment inundated with razor wire and cement, we are not simply in a state of arrested development. We need to be able to see people and things grow in that same

environment to remind ourselves that we can experience growth, despite our environment. Many of us have found beauty in weeds and flowers growing through the cracks in the pavement. There is both beauty and inspiration in knowing that we, who have fallen through the proverbial cracks in the system, can, if properly motivated and cultivated, grow through those very cracks.

Another participant described an instance in which ORT members, who knew that he was grieving his father's death, took him to the healing garden so that he could plant something in remembrance of his father:

And so on the day before my dad's year of being dead. . . an anniversary–they [ORT] took me out to the healing garden and said, "You should plant your dad a plant." I grabbed a fern, just because something about the fern was like my dad: strong as can be. As I'm planting this fern, the Resource Team was telling me good things and not to focus on the sad, depressive stuff that would probably get me in a place where I don't want to be. So, I do thank the Resource Team because they've been there for me every step of the way. When I had the Resource Team around me, they've been not letting go. And that's what people need, especially back here [in the BHU].

Other participants described the ways in which their newly developed rapport with ORT staff actually helped them establish better relationships with their clinical counselors. For instance, one participant credited the ORT with teaching him new coping skills, which he felt added to those his mental healthcare clinicians had helped him develop.

**Fostering hope and resilience among incarcerated people.** Many participants reported that the ORT had facilitated positive change and resilience. One man who had been incarcerated for decades credited the ORT with "opening a door that had been closed for decades. . . giving me an opportunity to deconstruct, rethink, and transcend the story that has dominated our lives." Another participant told us that the ORT had helped him and others to find healthier ways to process childhood trauma and the turmoil created by imprisonment:

Whether it applies to drug addiction, alcoholism, or just brokenness, the Resource Team helps us to do away with old ways and ideas of the things that are defeating our purpose in life. They help us overcome those things damaging us, to put into perspective that there is a way out of the cage.

One participant indicated that his newly developed relationships with ORT staff instilled a sense of hope for the future that had been lacking:

The Resource Team is helping me. I'm not just sitting in a cell sad, forlorn, broke down. I never, ever thought I would be working with guards, that I would be trusted at the level I'm trusted. They're building me up, so that gives me hope for the future.

Many participants said that the ORT helped them gradually overcome their anxiety about having social contact and reduced their tendency to self-isolate. An individual who had spent seven years in solitary confinement noted that having the option to join group activities and eat meals with ORT staff helped him feel more comfortable and less anxious in social situations, in ways that he felt would likely carry over into other settings:

I've been coming out [of my cell] for a lot more yards, a lot more day room [time], and learning how to be around people so when I go into society it's not such a culture shock. Because when I got out of federal prison doing seven years [in solitary confinement], when cars would drive by me or big trucks on the sidewalk, I would get defensive and uncomfortable and break a sweat, a cold sweat. I had a phobia of the public. I was diagnosed with it. Besides family and friends, I don't normally talk to strangers or people. But when it comes to the Resource Team at OSP, I get fired up.

Others described ways in which they felt social interactions with the ORT helped alleviate some of their mental health symptoms, including depression and even hallucinations. Activities such as playing basketball or dominoes with ORT staff and peer mentors provided opportunities to talk through past traumas in their lives, resolve prior conflicts with other staff and incarcerated people, and work on ways to counteract feelings of despair. One participant said that, prior to working with the Resource Team, he had little hope in ever getting out of long-term isolation. He said, "the Resource Team does not take the approach of locking you up and throwing away the key because you're through. Instead, they say, "No, you're not through. We will see you through." This theme—that the ORT's persistence in offering help, no matter how many refusals they faced, was voiced by many. As one participant put it:

They're just loyal. That's what the Resource Team is around here. You know how many times I tried to tell them, "Don't show up at my door. I'm not going." They still showed up. "How you doing, man? Come on, man. Let's go play basketball, man. You got to get out your cell." I said, "Why? I'm not even in the mood." And then I'll go. . . and I'm telling you, I come back in a better mood, more focused on what I need to do. And that's because they don't give up. They're not going to give up on you, because if you got everybody giving up on you, that's how the hell you end up in prison.

As an innovative extension of the Norwegian RT model, the ORT recruited several incarcerated persons who had experienced solitary confinement and led peer-support programs to serve as "peer mentors" for the ORT program participants. The program recruited peer mentors who were respected by other incarcerated persons and trusted by staff, making them credible messengers for educating others about the objectives and potential benefits of the program. These men provided an additional layer of emotional support to participants by joining them in social activities and helping them establish and work toward their personal goals. In fact, a number of participants emphasized that having a peer mentor was an essential part of the program. They acknowledged that peer mentors did many of the same things ORT staff members did—participating in social activities with them, supporting their progress, and alleviating social anxieties—but also noted that, as peers, they could provide certain insights and guidance that correctional staff could not. For example, one participant said:

In the Qu'ran they say that there's people you should follow, and there's people that will lead you, and there's people that will definitely take you down. But he's [peer mentor] the one that leads me. He's like the Resource Team. . . but with him not being police, he's one of the main people that inmates [sic] should go to because if it wasn't for him, I probably would've punched a lot more people and been in a whole bunch of situations.

**The ORT instilled a sense of safety and reduced feelings of vulnerability among participants.** An analysis of qualitative interviews, administrative data, and ORT logs suggested that

the ORT helped to instill a sense of safety and stability among its participants. This, in turn, helped to reduce the distrust and volatility that had permeated the BHU in the past, often escalating into self-harm, cell extractions, and extended solitary confinement. Incarcerated participants and staff alike were acutely aware of the dramatic reduction in use-of-force incidents (documented above), and they described a corresponding change in staff demeanor. As one incarcerated participant observed: "The approach is different (now), even from security. . . The lieutenants now are addressing things instead of just gearing up . . . they're taking a more time-centered approach . . . giving space and reasoning with the guys. Cell extractions [were frequent] and now. . .When was the last one? I can't even think of it." Another ORT participant stated: "The Resource Team has helped me take a new perspective. Now, I look at violence [as] a cowardly thing versus a brave thing and that unless I'm defending my country, family, or myself, there's no reason you utilize it."

Participants acknowledged that the improved social dynamics between incarcerated persons and ORT members helped to reduce tensions in the unit, made the de-escalation of conflict more likely, and reduced the frequency of violence and injuries. For example, one participant explained that during his time in a mental health unit at another prison, he was in a constant state of "fight or flight" in his interactions with officers. He said "at [those prisons], I got aggressive, and I fought with staff" because he felt disrespected, mistreated, or dehumanized by people with control over nearly every aspect of his life. He explained that, since working with the ORT, "I don't get those feelings. I don't get those urges" and that the ORT has helped him come to view officers in a different light.

Another participant remarked: "[ORT members] give you some space, reengage with you when you're safe, and then move forward. It's making this experience [incarceration] a lot more civil and a lot less barbaric and brutal. I think it helps the staff and [incarcerated people] connect instead of feeling like enemies." Others described the ORT as reducing the animosity that typically exists between incarcerated persons and staff.

> When I look at [the Resource Team], I don't see police that wants to spray me in the face. I see people that I can just talk to if I'm having a bad day. So I see him as a human versus an officer, and I feel like they view me as a human versus a prisoner. So that creates that dynamic to where you can actually have a legitimate conversation without having an adversarial position. That goes a long way. And they're here to support me, so that bond is unbreakable.

Interestingly, both the incarcerated persons and ORT staff suggested that the reduced uses-of-force by officers had helped increase their feelings of safety in the environment. Overall, most participants said they felt less vulnerable and some even said the change in atmosphere reduced their need to engage in self-harm. Administrative data bore this out: 81.8% of participants who had repeatedly engaged in self-injurious behavior and who had been subjected to use-of-force (e.g. cell extractions, chemical spray) before engaging in the ORT were no longer involved in these kinds of encounters.

**Case examples of decreases in staff use of force due to the ORT.**   When we looked more specifically at the nature of staff uses-of-force, we realized that they had decreased most against those persons who had been its most frequent targets in the past. For example, one incarcerated person accounted for a disproportionately high number of use of force incidents from 2016 until his enrollment in the ORT. This participant, a man in his early forties, had spent 73% of his incarceration in solitary confinement. Before his first interaction with the Resource Team, he had received 20 incident reports for staff assaults (frequently involving throwing bodily substances), self-injury, and property damage (e.g., flooding of cell). Staff had responded with uses

of force against him in fully 80% of those incidents (e.g., cell extractions and/or chemical spray). Nearly half of them (43.8%) had resulted in bodily injuries that required triage to a clinic or hospital for medical treatment. The ORT began working with him in March 2021 and, by the end of that year, had engaged with him in 25 weeks of individually planned activities outside his cell. Over that time span, his mean rate of disciplinary infractions decreased by 54.3% (from 3.15 to 1.44 per 100 incarceration days) compared to the years before enrollment in the program; and his mean rate of assaults decreased by 40.8% (from 0.49 to 0.29 per 100 incarceration days). Because many staff uses-of-force occurred in response to instances of self-injury, it was especially important that this participant did not have any recorded self-injuries since his enrollment in the ORT and, although he received five disciplinary infractions (including one staff assault), none of those incidents precipitated a use-of-force.

## Impact of the Oregon Resource Team on the health and well-being of correctional staff

We were also interested in the impact of the ORT on the health and well-being of the staff and addressed these issues in our interviews. Frontline correctional staff have immense influence as to whether legal reforms succeed or fail, and the attitudes and beliefs of correctional officers are crucial to the implementation of long-term correctional reform [38]. Fortunately, as we illustrate below, staff members overall also viewed the ORT in extremely positive terms, as an intervention that not only helped them to reduce the work-related stress, conflict, and violence they had previously experienced in the BHU but also enhanced their job satisfaction and sense of occupational purpose.

**Recognizing the mutual harmfulness of solitary confinement.** In the course of our interviews, many officers reflected on the ways they had been trained to respond when BHU residents engaged in violent or disruptive behaviors (including the use of cell extractions, restraints, physical force, chemical spray, and increased isolation), noting that these responses were often ineffective and even counter-productive (ultimately increasing contentious interactions with incarcerated persons and engendering more assaultive behavior). As they also noted, encounters involving use-of-force adversely affected their own mental health. One staff member explained that, before the implementation of the ORT, "there was always this cloud over this [special housing wing]; the first few years I was [in BHU] this was a horrible place to work and just a negative place to be." One officer likened the environment to his combat experience in the military:

> I've seen this place turn a lot of officers into different people. It's 95 to 98% mundane, routine, and then 2 to 5% absolute chaos. When you work in a place of despair for that long . . .to tell you the honest to God truth, it reminded me of war. And when you live in a constant arena of that, it does things to you.

Other officers acknowledged that the stressfulness of working in such a chaotic, potentially dangerous environment threatened to spill over into their personal lives. For example, one officer recalled feeling "drained" from constantly having to "make a concerted effort not to take all that negativity from work home to your family and cast it on to them."

Many ORT officers also acknowledged that they had gained insights into the harmfulness of solitary confinement on incarcerated people and its potentially detrimental ramifications for public safety. One officer lamented, "there are so many people languishing in a prison cell, that includes all of DOC, nationwide and worldwide. Sitting in the cell doesn't change anybody. And we've seen it here, most of the time it does the opposite, it makes them worse. So

why not try something new?" Some said that the goal of undoing these harms was what inspired them to participate in the ORT, and others were grateful to work in a more compelling, interesting, and varied occupational role, one that instilled a deeper sense of meaning in what they did. One staff member who had participated in the Norway immersion program and led the early efforts to develop the ORT, said:

> [This experience] gave me the guts and courage to act on what I knew deep in my heart was the right way to do things and not be afraid to move forward. One of the things that has come out of this is allowing staff to tell you how they feel and what feels right; because, at the end of the day we're all human, and we want to feel good about what we do, and we want to make other people feel good. Allowing staff to do that and talk about it is what we're doing.

**The impact on staff of reducing violence.**  ORT staff were well aware of the interplay between reductions in uses-of-force and reduced assaults in the BHU. Even during the diverse range of recreational, social, and leisure activities facilitated by the ORT, which required close contact with incarcerated persons with a history of violence, not a single incident of violence or major disruption had occurred during one of these activities. Staff attributed this in part to the innovative training they received from the Norwegian prison officers. As one sergeant said, "The Resource Team has really worked. We haven't had a staff assault yet. We know that it's always a possibility, but we haven't had that with the way we train and get to know each other."

Not surprisingly, then, ORT staff members credited the program with engendering a greater sense of safety in the BHU. When asked about the most beneficial aspects of the program, one prison official stated that "the number one [benefit] is that use of forces, staff injuries, staff assaults, sick leave usage, and self-harm incidents have all gone down."

ORT staff members shared this view. As one stated, "you can just feel it. And I think that's on both sides, adults in custody and the staff. Everybody feels better, it's relieving everybody's tension, everybody's stress is going down a little bit, which is making it a better environment."

Many staff acknowledged that in the past they had been quick to deploy force against residents in special housing units. Several reported having themselves experienced injuries during "cell extractions" that required medical attention and sick leave, and/or having been assaulted by residents during cell-front encounters or on the tiers. Many also reported that the ongoing exposure to such violence had resulted in stress, hypervigilance, and fear for their own safety. At times, they felt "emotionally numb" to persons who were deteriorating mentally and to the experience of taking part in behavior that resulted in injuries and psychological harm. They credited the dramatic reductions in violent incidents as instilling a greater sense of safety, reduced stress, and helping to create an environment that was more conducive to responding to residents' legitimate needs in caring rather than oppressive ways.

**Enhanced job satisfaction and wellness.**  The ORT training had provided officers with the knowledge, skills, and inspiration needed to improve quality of life for the incarcerated persons under their supervision and they reported that this, in turn, had greatly improved their job satisfaction. One sergeant said that "after working three terms in special housing. I swore I would never go back, because it rips the soul out of the body." However, working the ORT had changed his perspective:

> Now, with the resource team. . . we do not sit in the mess of the same thing every day. When we get to interact with [the residents], we get that one-on-one time. We get to see a

different person when we can get them out of that soul-depleting environment and see the positive in them too. There's a remarkable change in their face, their shoulders straighten up. . . and so this [Resource Team] has provided an outlet to help some of these guys take a leap forward, and it's been exciting to see this change.

Correctional staff also commented that participation in the ORT allowed them to engage more thoughtfully and creatively with residents, proactively tailoring their interactions to the person's individual interests and concerns. Relatedly, staff reported that their participation on the ORT had enlarged and enhanced their occupational role, empowering them to operate outside the rigid policies they had previously been required to follow. According to one staff member: "Now we treat every situation as an isolated individual situation. With the individuals involved, we don't have a cookie cutter approach anymore. Instead, it is 'this person John Smith. What is John Smith's motivators, what's his past history, who does he like.'"

Many staff reported that although their occupational roles had in certain ways gotten more complex, the enhanced rapport and communication they now had with residents actually made the job less stressful. As one remarked, "This is obviously a very stressful job. There are still a lot of stressors, but I feel better, less stressed. It's caused me to do some analyzation of myself and where I'm at in life. It's opened my eyes to how to be a better person." Others reported that their participation on the ORT was actually transformative, an experience that had improved their job satisfaction and cultivated personal wellness. A sergeant who had worked at OSP for over a decade recalled, "I was widely known around here for a long time as being Mr. Negative. That was kind of my trademark. And this [joining ORT] was the best thing that ever happened to me because a lot of my negativity has gone away." Another officer stated:

I've found a purpose. I feel like we're doing positive things here and seeing the results is amazing. [. . .] Making positive change. And what's important is we're making positive change for the [BHU residents], but in doing so, we're building staff wellness.

Correctional administrators also credited the Resource Team as contributing to the decreased use of sick leave among participating staff members. As one official explained: "Well, the last time we analyzed the numbers, we saw a 23% reduction in the use of sick leave. And we think that is definitely a reflection of how people feel."

## Dynamic security and the use of less restrictive conditions

Prior to the implementation of the ORT pilot, a person's housing status in the ODOC (and therefore their potential placement in long-term isolation) was determined solely by use of a rigid security-focused classification system. Once a person was placed in specialized housing (such as the BHU), obtaining a transfer to general population required that they successfully complete a set of prison-mandated programs. Many seriously mentally ill persons simply could not manage to do this, which often resulted in extended stays in isolation. To alleviate this problem, the ORT introduced a process of dynamic, ongoing risk assessment that was based on the Norwegian correctional principle of "dynamic security" in which a correctional officer's knowledge of and interaction with an incarcerated person guides the appropriate level of security. As implemented in the ORT, residents who participated in out-of-cell activities and were able to successfully demonstrate and sustain the social skills required to live safely in a lower security setting were increasingly returned to general prison population housing units. One clinical staff member described the way that this approach better prepared residents for

life after the special housing unit: "Of all the guys that I've worked with [since the ORT] years, the majority of them haven't come back [to BHU]. And for the guys who have come back, they've come back for incidents that didn't rise to where they were three years ago."

One ORT officer recounted the plight of one chronically isolated person who suffered severe mental and physical deterioration during the many years he was in solitary confinement and described how the new, more humane approach helped alleviate his condition:

> We had one person that would scream literally 20 to 23 hours a day, bang his head. He had a gash on his head that was open all the time, running the risk of infection. Played with his feces on a daily basis. That individual is actually out doing field trips out into [the prison's] general population and getting ready to move to a more normal housing unit from the behavioral health unit. Prior to this program, people would tell me [the BHU] is where this person will live until he gets out.

Officers reported that being able to help alleviate the suffering of persons who had been harmed in these ways was personally rewarding. As one officer reflected, "[T]his program gives me the opportunity to help people and not hurt them. It is much better for me mentally to feel like I'm actually helping people instead of warehousing someone. The reality is that I'm now able to help these people so they can leave."

## Discussion

Solitary confinement adversely affects the incarcerated persons who experience it [2, 4] and the staff required to oversee the regime [12]. Yet available evidence indicates that it largely fails to deter violent or disruptive behavior, and may do more to imperil than promote safety both inside prison and after release [18, 39]. Balancing the known costs of solitary confinement against its few (if any) benefits has led some 40 states to pass laws to regulate and reduce solitary confinement since 2009 [40]. A growing number of professional organizations have urged reforms that would restrict if not completely eliminate the use of long-term solitary confinement [41]. However, as we noted at the outset of this article, the actual implementation of the reforms faces a number of institutional obstacles and has proven challenging. Moreover, very few of these attempts have been systematically assessed. That has resulted in a dearth of empirical data on whether and how solitary confinement reforms can be devised and implemented to humanely address the needs of persons subjected to it. This is especially true for the group of persons at perhaps the highest vulnerability of being placed and retained in isolation—those with a history of violent behavior who are diagnosed with an SMI.

The present study builds on small body of research underscoring benefits of educating frontline correctional staff on crisis-intervention and other humane approaches to responding to the needs of incarcerated persons with SMI [42, 43]. Results documented the numerous positive outcomes that occurred when one such program, based on the novel Norwegian Resource Team model, was implemented in a special housing unit in the Oregon State Prison. Like its Norwegian counterpart, the interdisciplinary ORT brought together custody staff and healthcare professionals to better respond to the needs of an especially vulnerable and challenging group of incarcerated persons, precisely the ones who are too often neglected and consigned to harsh forms of long-term isolated confinement. In an important sense, the Resource Team model operates to radically reconfigure the nature of the correctional officer role in these settings—from one of wielding forceful control to extending humane concern—and it seeks to transform the "culture of harm" that characterizes many solitary confinement units [44] into a culture of caring. Among other things, team members were trained to use de-escalation over

uses-of-force and to engage with incarcerated participants in meaningful activities that prioritized social interaction and provided emotional support.

The participants themselves reported a range of beneficial outcomes that they attributed to the ORT intervention. The outcomes included, on the part of the incarcerated participants, significant reductions in instances of self-harm and feelings of emotional distress, a greatly improved outlook and sense of well-being, and increased involvement in a host of meaningful out-of-cell activities and social interactions. The ORT staff also credited program participation with enhancing their overall job satisfaction and sense of well-being. They reported developing a deeper understanding of the interplay between trauma, mental illness, and prison conditions that, in turn, empowered them to make a positive difference in the lives of the incarcerated persons whom they oversaw. At the same time as these numerous out-of-cell interactions and activities were taking place, incarcerated persons and staff alike described experiencing the environment as much safer rather than more dangerous–a perception that was reflected in the institutional data. That is, uses-of-force by staff dramatically decreased, as did participants' disciplinary write-ups in general and for assaultive behavior in particular, as well as instances of self-harm.

It is important to emphasize that the ORT intervention focused on an especially vulnerable and challenging group of people in solitary confinement units, most of whom were not only seriously mentally ill but also had histories of violence against others. They are traditionally considered the most difficult to manage in prisons. The fact that these demonstrated positive outcomes were achieved with *this* particular group of incarcerated persons suggests that it may be generalizable to the much larger group of persons who are housed in solitary confinement. It is likely that they, too, could greatly benefit from this Norway-inspired approach, not only improving their well-being and hastening their pathway out of isolation but also improving the well-being of the larger group of staff with whom they routinely interact in these units.

In this regard, one of our most promising findings was that the ORT's officer-led individualized engagement created what might be characterized as a "virtuous circle" in an environment not known for such dynamics, one in which participants reported feeling more motivated to engage in mental health services, their behaviors improved, and officers reporting feeling less stress and frustration. These findings offer preliminary evidence for how the presence of a Resource Team model may contribute broadly to reducing, and perhaps ultimately ending, the use of solitary confinement that relies on isolation, deprivation, and forceful institutional control. In fact, we found evidence of another kind of virtuous circle expanding into the larger Oregon system and into other states' prison, such that word of the successful, humanizing ORT reforms being implemented at OSP spread to other facilities, and the beginning of a process by which they, too, will implement a Norway-inspired reconfiguration of their solitary confinement units. For example, the early successes of OSP's program paved the path for the launch of another Resource Team pilot program at the Snake River Correctional Institution in Eastern Oregon, and inspired officials in Washington state and California to create their own programs based on this Norwegian-inspired model. The expansion of this model into new correctional contexts has set the stage for additional studies to examine the potential of this approach to addressing some of the most vexing aspects of solitary confinement reform. In this way, these preliminary findings also have important implications for lawmakers, advocates, and correctional officials seeking programmatic alternatives to the use of solitary confinement, including for persons with serious mental illness.

The ORT program was not without limitations and opportunities for improvement. For one, true to its name, the "Resource Team" model is relatively resource intensive. Its five full-time staff members were able to provide individualized engagements to only one-third of the persons in the BHU over a nine-month period. In addition, participants often went several

weeks between ORT activities; those interviewed yearned for more frequent opportunities to interact with ORT staff and peer mentors. Obtaining additional resources and redistributing existing ones to increase the size, operations, and training of the ORT staff would be essential to realize the program's full potential to significantly reduce the facility's use of prolonged solitary confinement.

In addition, policymakers who wish to implement or "scale up" an ORT-type intervention must navigate potential barriers to reform posed by existing labor relations practices. In fact, notwithstanding its impressive successes, the ORT pilot program itself was temporarily stalled due to labor practices that prioritized positions on the ORT by virtue of seniority and scheduling preferences rather than officers' demonstrated knowledge, skills, and commitment to the mission of the program. Overcoming such barriers may depend on policymakers' ability to document the benefits of ORT reforms to the health and safety of correctional staff and the creation of opportunities to enhance the professional skills of correctional staff will be essential to achieving buy-in from labor organizations [28, 45–47].

Recently, an organization of correctional officer union representatives called for an urgent response to the growing occupational health crisis among correctional officer in the U.S. [48]. As we have tried to show, the Resource Team simultaneously improved working environments for correctional officers, and enhanced the health and well-being of all concerned. In the final analysis, educating, engaging, and motivating frontline correctional staff to be a part of the process by which solitary confinement is so radically reformed may be essential to significantly curtailing its overall use and perhaps to its ultimate elimination. Such approaches may help foster alignment between advocacy groups and corrections stakeholders to find the legislative solutions needed to profoundly transform, and eventually end, the harmful practice of solitary confinement.

Indeed, bringing an ORT-type intervention to scale without increasing the overall size of the correctional workforce or further investing in carceral infrastructure will require correctional officials to adopt additional changes that drastically shrink the total number of people held in various form of solitary confinement. Examples might include significantly revising disciplinary policies to permit the use of solitary confinement only as a last resort, in response to only the most serious disciplinary infractions, and for the shortest amount of time possible [4]. Those persons previously housed for longer terms in solitary confinement could be diverted to more units repurposed to provide specialized treatment, particularly mental healthcare. As correctional agencies face the challenge of implementing legislation and other types of legal reforms, it is essential for state actors to recognize that incarcerated individuals who are deemed to require (or desire) physical separation from others due to serious health and safety reasons, need an environment led by compassionate professionals who are skilled at fostering meaningful social relationships, promoting engagement in clinical care, and creating professional interactions that are empathetic and supportive of people with complex mental health conditions shaped by violence, trauma, and socioeconomic adversities.

Undoing "cultures of harm" and creating cultures of caring will also require authorities to critically identify and rescind existing policies that counteract the ethos and objectives of reform, such as those that implicitly authorize frontline officers to dehumanize incarcerated persons through deprivations of essential social and material resources, reliance on static security measures (e.g., restraints) that often result in psychological and bodily harms. Academic researchers working within the walls of correctional systems may turn to established theories of dehumanization and "moral disengagement" used to assess the structural forces, institutional policies, and interpersonal factors that shape state actors involvement in perpetuating human rights violations to create frameworks for critiquing exiting policies, understanding dynamics between officers and incarcerated persons, the range of harms solitary confinement

inflicts, and creating organizational contexts to align solitary confinement reforms with international standards [4, 26, 49, 50].

In addition, reaching the full potential of the ORT model, in Oregon and other prison systems, may depend on the ability to achieve the goal of decarceration more generally—not only downsizing the prison system overall but also reducing the nation's overreliance on carceral responses to crime and replacing it with more humane, social justice-oriented, public health-based approaches [51]. The ORT's transformation of a traditional solitary confinement unit from a culture of harm to a culture of caring conveys a broader message about what can be achieved when institutional systems are premised on recognizing the inherent dignity and humanity of the persons who are confined in them as well as those who work there. While shrinking prison populations mostly falls outside their legal powers, correctional staff can leverage their influence to call for system-level population reductions as means to foster safer and less stressful work environments that afford more opportunities for correctional staff to provide meaningful professional services and social support to meet the needs of people in their care and custody.

Finally, with this latter point in mind, we fully acknowledge that, despite the fundamentally different nature of the interpersonal relationships the ORT program engendered between staff and incarcerated participants, the meaningful interactions that were fostered between them, and the extraordinarily positive changes that both groups attributed to this intervention, it nonetheless operated within a traditional carceral setting in which stark power differentials, numerous limitations to personal liberty and forms of dehumanization necessarily and insurmountably remained. The Norwegian goal of "normalization" notwithstanding, the ORT sought to better approximate but certainly could not remotely replicate normal living conditions or interpersonal relations. Thus, it represents a significant reform in what we hope is a promising beginning in an eventual progression of further reforms that eventually will result in the practice of solitary confinement being replaced entirely by more humane alternatives.

Moreover, this study, and its promising findings, should be considered in light of several limitations. First, it is possible that the extremely positive outcomes we documented were a product of the persons involved rather than the program they implemented. Although we see the two as inseparable, we acknowledge that the highly motivated, dedicated ORT staff may be difficult to replicate. Similarly, the prisoner participants who volunteered to participate may not be representative of the overall population targeted by this reform. Second, although ORT participants reported improvements in their health and well-being and attributed them to the ORT, a descriptive, qualitative case study cannot allow us to draw definitive causal conclusions. We adopted a case-study design to detail the ORT for its utility in describing the development and early implementation of pilot programs allowing us to set the stage for future evaluations as the program increases in scale.

Nonetheless, we believe that the results from this preliminary case-study provide important insights into participant perspectives on why and how this kind of intervention can help to address the complex needs of an incarcerated population experienced as especially challenging by custody staff and provide a more humane, person-focused approach to reducing the use of long-term solitary confinement. As the ORT model evolves and expands to new settings and populations, quasi-experimental and comparative prospective study designs using repeated assessments of health outcomes will strengthen evidence about its impact.

## Conclusion

Limiting if not eliminating the harmfulness of solitary confinement for people with SMI and other incarcerated persons has become a central focus of litigation, legislation, and corrections

department-initiated reforms [1, 52, 53]. As the present case study shows, the Norwegian Resource Team model, adapted for use in an Oregon prison, delivered very promising early results. Participants reported that significantly changing the nature of the correctional officer role and reducing the level of isolation to which residents were subjected improved their health and well-being and reduced violence (assaults and staff use of force). The positive impacts observed in this case study suggest likely gains from reallocating resources away from correctional practices that reinforce the costly, inhumane and harmful reliance on solitary confinement, and towards those that center human dignity, public health, and social justice.

## Author Contributions

**Conceptualization:** David H. Cloud, Craig Haney, Brie Williams.

**Data curation:** David H. Cloud, Dallas Augustine.

**Formal analysis:** David H. Cloud.

**Funding acquisition:** Brie Williams.

**Investigation:** David H. Cloud.

**Methodology:** David H. Cloud.

**Writing – original draft:** David H. Cloud, Dallas Augustine, Brie Williams.

**Writing – review & editing:** David H. Cloud, Craig Haney, Dallas Augustine, Cyrus Ahalt, Brie Williams.

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
