## [Decision Letter · Decision Letter 0]

14 Mar 2023

PONE-D-23-00201The Resource Team: a Case Study of Solitary Confinement Reform in OregonPLOS ONE

Dear Dr. Cloud,

Thank you for submitting your manuscript to PLOS ONE. After careful consideration, we feel that it has merit but does not fully meet PLOS ONE’s publication criteria as it currently stands. Therefore, we invite you to submit a revised version of the manuscript that addresses the points raised during the review process.

We would be grateful if you could submit your revised manuscript by Apr 28 2023 11:59PM. If you will need more time than this to complete your revisions, please do not hesitate to request for an extension by replying to this message or contacting the journal office at plosone@plos.org. Please include the following items when submitting your revised manuscript:A rebuttal letter that responds to each point raised by the academic editor and reviewer(s). You should upload this letter as a separate file labeled 'Response to Reviewers'.A marked-up copy of your manuscript that highlights changes made to the original version. You should upload this as a separate file labeled 'Revised Manuscript with Track Changes'.An unmarked version of your revised paper without tracked changes. You should upload this as a separate file labeled 'Manuscript'.

We look forward to receiving your revised manuscript.

Kind regards,

Dr Nasrul Ismail

Academic Editor

PLOS ONE

Journal Requirements:

"This research was supported by the Jacob and Valeria Langeloth Foundation, The Charles and Lynn Schusterman Family Philanthropies, Arnold Ventures, and other private donors."    

"Authors DC, DA, CA, and BW are members of the Amend team at UCSF. CA, BW, and CH co-facilitated the Norwegian exchange program and provided the technical assistance to develop the Resource Team intervention."

Additional Editor Comments:

Please note that the PLOS Data policy requires authors to make all data underlying the findings described in their manuscript fully available without restriction, with rare exception (please refer to the Data Availability Statement in the manuscript PDF file). The data should be provided as part of the manuscript or its supporting information, or deposited to a public repository. For example, in addition to summary statistics, the data points behind means, medians and variance measures should be available. If there are restrictions on publicly sharing data—e.g. participant privacy or use of data from a third party—those must be specified.

Reviewers' comments:

Reviewer's Responses to Questions

**Comments to the Author**

1. Is the manuscript technically sound, and do the data support the conclusions?

Reviewer #1: Yes

Reviewer #2: Partly

Reviewer #3: Yes

2. Has the statistical analysis been performed appropriately and rigorously? 

Reviewer #1: Yes

Reviewer #2: No

Reviewer #3: Yes

3. Have the authors made all data underlying the findings in their manuscript fully available?

Reviewer #1: No

Reviewer #2: No

Reviewer #3: Yes

4. Is the manuscript presented in an intelligible fashion and written in standard English?

Reviewer #1: Yes

Reviewer #2: Yes

Reviewer #3: Yes

5. Review Comments to the Author

Reviewer #1: This paper describes an intervention related to solitary confinement practices in the Oregon state prison system called the “Oregon Response Team”. Triangulating across interviews and administrative data, the authors provide both qualitative and quantitative evidence of the impacts of the intervention for both incarcerated people and correctional staff. A particular strength of the paper are these two facets of the analysis: that the data draw on both incarcerated people and staff, and the interviews and administrative data analysis complement one another.

Below, I detail a few ways I believe the contribution and limitations of the study could be further underlined:

Background:

The paper could be improved with a more extensive discussion of other types of interventions to reduce/end solitary confinement. While the authors note that there have been limited evaluations, it would be helpful to know what those studies found, and how this type of intervention resembles or differs from other kinds of reforms (litigation-driven, DOJ, legislation, DOC-initiated restrictions on length of stay). For example, some reforms focus on particularly marginalized groups, while others focus on the practice as a whole. Would this model if replicated be impactful for an entire system population, or just those who are designated eligible?

Data/Methods:

Could the authors provide more context for the interviews? What specific questions were asked (e.g., could they append the interview protocol)? As the authors note, this article is meant to describe the program, not offer causal evidence of its effects. Nevertheless, it would be useful to understand why this particular method was chosen, and how future studies could assess the impacts of the program (for example, longitudinal design, comparision groups).

Implications:

As the authors note, this is an intensive program requiring many resources to replicate and scale. Could the authors provide more detail on how they think this kind of intervention could be replicated or scaled across several prisons or prison systems? Would this work in all kinds of correctional settings where solitary confinement is used? What are the conditions that could make this program more accessible?

Reviewer #2: Solitary confinement has been used as a form of judicially sanctioned punishment since the 1820s. Studies highlighting the negative health consequences associated with the use of isolation as a means of managing prisoners were first published in the 1840s. Despite continuing adverse reports from clinicians, psychologists, psychiatrists and non-governmental organisations highlighting the risks associated with limiting opportunities for prisoners to socialize with others, the solitary is widespread across many correctional systems. This is especially true of the US.

This submission reports on a recent attempt to implement a Norwegian Correctional Service initiative in the Oregon State Penitentiary, a maximum-security institution. This initiative was designed to improve rehabilitative outcomes for high-risk prisoners—especially those with serious mental illness (SRI). The program was limited to the Behavioral Health Unit (“BHU”) at the Oregon State Penitentiary. The inmates of the BHU have ‘serious and lengthy mental health histories and past instances of violence against themselves or others’. The case study describes and evaluates the work of the Oregon Resource Team (ORT) over a nine-month period in 2021.

The number of participants in the study is small. While 44 inmates took part in the ORT trial over the course of the study period, only16 were interviewed. Administrative data could be accessed for all 44. These were supplemented with interviews with 14 correctional staff. The study uses this data to plot quarterly unit-level changes in staff uses of force from January 1, 2016 to December 31, 2021, a time period that includes the period before the implementation of the ORT experiment and its first 12 months of operation.

Although the study purports to report on the impact of the ORT experiment on the ‘behavior, health, and well-being of incarcerated persons and correctional staff, and ways to optimize its effectiveness and expand its use’ its major quantitative findings relate to falling levels of violence. Qualitative evidence from interviews is used to assess other positive impacts on the well-being of inmates and (interestingly) staff.

It would be particularly useful if the window in which the ORT experiment was implemented was marked on Figure 1. Even so it is apparent that the peak in use of force was in the last quarter of 2018 and that considerable amount of the reduction occurred in 2019 and especially 2020. Perhaps Table 2 could be reorganised to show the change in the use of force between years (eg. 2016, 17, 18, 19, 20 and 21 – the last being the year of the study). More needs to be said about selection into the BHU over this time longer time frame and its operation. More needs to be said about the selection of individuals into the study. It might also be an idea to compare the ORT experiment with the outcomes of its Norwegian counterpart. Were they similar? A great deal of the submission is taken up with analysis of the interviews. These are informative but lengthy.

This is an important study which could lead to significant reforms in current penal management but it needs to make sure that the statistical findings are robust and then use the qualitative evidence to flesh these out.

Reviewer #3: This is a spectacular piece of scholarship that I am excited to see (hopefully) published. I have very little in terms of edits as it is already well-written and rigorously presented. Just a couple of very small details to consider

1. At several places the authors use numerals rather than spelling out numbers. I'm not sure of the rules/guidelines for this journal but standard APA practice is to spell out numerals (not %) from zero to 11 and then use numerals for the rest. So 3, would be three. Totally minor, and a bit nitpicky. I apologize.

2. The literature review is fairly comprehensive but seems to be missing a new book that came out last year: Surviving Solitary by Rudes and colleagues. I'd consider working that in since the main message is about harm. I did see the citation of several pieces by Reiter though, and that's the other biggie to cite (in my view). Kudos.

Thank you for your work and your passion for it. Loved reading this piece.

6. PLOS authors have the option to publish the peer review history of their article (what does this mean?). If published, this will include your full peer review and any attached files.

Reviewer #1: No

Reviewer #2: **Yes: **Hamish Maxwell-Stewart

Reviewer #3: No

---

## [Author Response · Author response to Decision Letter 0]

17 May 2023

Reviewers' comments and authors' responses:

Comments to the Author

1. Is the manuscript technically sound, and do the data support the conclusions?

Reviewer #1: Yes

Reviewer #2: Partly

Reviewer #3: Yes

2. Has the statistical analysis been performed appropriately and rigorously?

Reviewer #1: Yes

Reviewer #2: No

Reviewer #3: Yes

3. Have the authors made all data underlying the findings in their manuscript fully available?

Reviewer #1: No

Reviewer #2: No

Reviewer #3: Yes

We understand and value the journal’s commitment to data transparency and sharing. However, we are unable to upload either the quantitative or qualitative data that we used for this case study, due to legal restrictions set by data sharing agreement between UCSF and the Oregon Department of Corrections. These datasets contain sensitive information for incarcerated people, and our protocol was designed to protect the privacy of this medically and socially vulnerable group. For other scholars who are interested in accessing these data, we recommend contacting the Oregon Department of Corrections’ Research Committee. We would be glad to take advice from reviewers about other approaches that would allow us to meet our IRB requirements, and the legal requirements of our data use agreement with the department of corrections around sensitive data while complying with the data transparency and sharing policies of the journal. 

We have revised our data availability statement, which reads as follows:

o “Data Availability: This study utilized both administrative and qualitative data. Administrative data cannot be shared publicly because the administrative data we analyze in this paper is drawn from a confidential data file, shared with the research team for the limited purpose of evaluating the Resource Team pilot in the Oregon department of corrections. If any researchers wish to obtain similar data from the Oregon department of corrections, the authors of this paper would be willing to consult with those researchers about the request and the process for obtaining the data. In theory, the administrative data file used in this study could be accessed again by future researchers. External researchers can also access the Oregon Department of Corrections Research Committee form on the agency’s website (https://www.oregon.gov/doc/Forms/cd-1838-research-application.pdf) to learn about the legal requirements and application process.

We are also precluded from publicly sharing qualitative data. These transcripts contain sensitive information for correctional staff and incarcerated people, and our protocol was designed to protect the privacy of this medically and socially vulnerable group. Providing additional information beyond anonymized quotations reported in the findings (such as full transcripts or data sets) would enable identification of the participants, especially when combined with administrative data. This would breach participants’ confidentiality under terms of informed consent and protocol of the University of California Institutional Review Board, and data agreements with the Oregon Department of Corrections Research Committee.” 

4. Is the manuscript presented in an intelligible fashion and written in standard English?

Reviewer #1: Yes

Reviewer #2: Yes

Reviewer #3: Yes

5. Review Comments to the Author

Reviewer #1: 

a) This paper describes an intervention related to solitary confinement practices in the Oregon state prison system called the “Oregon Response Team”. Triangulating across interviews and administrative data, the authors provide both qualitative and quantitative evidence of the impacts of the intervention for both incarcerated people and correctional staff. A particular strength of the paper are these two facets of the analysis: that the data draw on both incarcerated people and staff, and the interviews and administrative data analysis complement one another.

Below, I detail a few ways I believe the contribution and limitations of the study could be further underlined:

Background:

The paper could be improved with a more extensive discussion of other types of interventions to reduce/end solitary confinement. While the authors note that there have been limited evaluations, it would be helpful to know what those studies found, and how this type of intervention resembles or differs from other kinds of reforms (litigation-driven, DOJ, legislation, DOC-initiated restrictions on length of stay). For example, some reforms focus on particularly marginalized groups, while others focus on the practice as a whole. Would this model if replicated be impactful for an entire system population, or just those who are designated eligible?

Thank you for this suggestion. We have added a couple of recent examples of solitary confinement reforms to our background and also added language to explain how this intervention fits into various approaches to reform. We now also include a brief discussion of the factors leading up to the Resource Team program, which included the threat of litigation and terms from a memorandum of understanding between the Oregon Department of Corrections and Disability Rights-Oregon to reduce the use of solitary confinement in the Behavioral Health Unit. We view the resource team approach as part of a multi-faceted approach, but also one that aims to inspire the front-line officers to play an active role in positive change, rather than directly or indirectly opposing reforms that are mandated from above as a result of litigation or legislation. We have aimed to emphasize that there are few empirical studies looking at the impacts of solitary reform efforts, including department initiated and legislatively mandated approaches. 

We have revised the background section as follows:

“People with serious mental illness (“SMI”) are disproportionately exposed to and harmed by this penological practice (5, 6). Indeed, preventing exposures to solitary confinement for people with SMI is a central focus of advocacy, litigation, legislation, and system-initiated reforms. Yet, even when efforts to adopt new laws and policies designed to restrict the use of solitary confinement are successful, they often face a long and challenging implementation process before bringing about transformative change. A complex implementation process must follow to translate new policies into practice. Despite widespread calls to significantly restrict, reform, or end the use of solitary confinement (7, 8), correctional agencies have struggled to heed them, even in jurisdictions with correctional officials more receptive to change(9-11). A variety of structural, legal, organizational, and cultural factors may influence whether and how solitary confinement reform efforts succeed in achieving their objectives. 

Because only a few attempts to significantly reform the practice have been empirically assessed (9, 10, 12), additional studies are needed to build upon an important, emerging body of scholarship focused on assessing whether and how reform efforts achieve their intended objectives. For example, one case-study unpacked the policy changes that North Dakota correctional leaders initiated, which achieved significant reductions in the use of solitary confinement, and were credited with decreasing violence and other benefits for incarcerated persons and staff alike (12). Schlanger (2020) utilized an “incrementalist versus maximalist” framework to identify the policy levers and institutional factors that may facilitate or impede reforms from achieving their stated goals (11). Similarly, Augustine et al. (2021) reported on successes and limitations of initiatives to improve conditions within solitary confinement units in Washington prisons (9). The present study builds on this emerging body of scholarship focused on assessing whether and how reform efforts achieve their intended objectives. 

b) Data/Methods:

Could the authors provide more context for the interviews? What specific questions were asked (e.g., could they append the interview protocol)? 

We have added several sentences in the methods to provide more detail on the types of questions that participants were asked during interviews. 

This section now reads:

• We conducted semi-structured interviews with subsamples of the incarcerated persons who participated in the program (interviewed in July 2021) and the staff who devised and implemented it (interviewed in May 2021). Due to COVID-19 related travel restrictions, we were only able to visit the prison to recruit participants for qualitative interviews for two days in July 2021. Of the 44 persons who participated in the ORT during the study period, 38 had begun participating by the time of our July 16, 2021, interviews. Of the 38, 17 had been moved out of the housing units or were not available at the time of our interviews. In addition, as a measure of the degree of vulnerability and impairment of the population of persons who participated in the ORT, four were deemed by healthcare professionals to be too impaired to consent for in-person interviews. Of the remaining 17 ORT participants, one person declined an interview, resulting in a total of 16 ORT participant interviews. 

Our semi-structured interview format focused on participants’ experiences before and during incarceration, the amount of time spent in solitary confinement, and their experiences with the ORT. Researchers used an interview guide with a series of prompts intended to elicit participants’ lived experiences with solitary confinement (e.g. frequencies of admission, reasons for placement in solitary confinement, lengths of stay, access to programming, recreation, treatment etc.) and how those experiences affected their overall well-being. Participants were also asked about their initial and ongoing interactions with the Oregon Resource Team members, and whether and how the program had affected them personally, as well as their experience of the climate and culture in the special housing units. They were also asked to discuss any positive aspects of the program and give feedback on ways to enhance it. All recruitment, consenting, and interviews were conducted in a private office, where no correctional staff were present. For interviews with incarcerated persons, we obtained verbal consent to further protect confidentiality, which was documented by the interviewer. All participants were unrestrained during the interviews, which lasted 60-90 minutes and were audio recorded.

c) As the authors note, this article is meant to describe the program, not offer causal evidence of its effects. Nevertheless, it would be useful to understand why this particular method was chosen, and how future studies could assess the impacts of the program (for example, longitudinal design, comparison groups).

Thank you for this suggestion. We have added language to the discussion section to address this point. We also added examples of questions from the interview guide, and examples of prompts in the guide. This case study was formative and has provided important insights into how to enhance the evaluation of similar Resource Team programs in the future, such as via longitudinal cohort studies and propensity score matching and other quasi-experimental methods to allow for comparisons. Given the relatively small scale of this program and its evolution during its pilot stages, the approach we took was well suited for this study’s objectives. 

We have edited the discussion section to reflect these points as follows:

• “Second, although ORT participants reported improvements in their health and well-being and attributed them to the ORT, a descriptive, qualitative case study cannot allow us to draw definitive causal conclusions. We adopted a case-study design to detail the ORT for its utility in describing the development and early implementation of pilot programs allowing us to set the stage for future evaluations as the program increases in scale. Nonetheless, we believe that the results from this preliminary case-study provide important insights into participant perspectives on why and how this kind of intervention can help to address the complex needs of an incarcerated population experienced as especially challenging by custody staff and provide a humane, person-focused approach to reducing the use of long-term solitary confinement. As the ORT model evolves and expands to new settings and populations, quasi-experimental and comparative prospective study designs using repeated assessments of health outcomes will strengthen evidence about its impact.”

d) Implications: As the authors note, this is an intensive program requiring many resources to replicate and scale. Could the authors provide more detail on how they think this kind of intervention could be replicated or scaled across several prisons or prison systems? Would this work in all kinds of correctional settings where solitary confinement is used? What are the conditions that could make this program more accessible?

We agree that more detail about the scalability of this high-resource program would benefit readers. We have added a few sentences to the discussion section to reflect on scalability and replicability. We have also updated the manuscript to disclose changes that occurred with the Resource Team program at this facility that temporarily stalled its operations, its recent re-start, and what considerations are needed to enhance its sustainability. We also reference two new states that have adopted this model (which have not yet been evaluated) which provides evidence of scalability despite the resource intensive nature of the program. 

This section of the discussion now reads as follows:

“Indeed, bringing an ORT-type intervention to scale without increasing the overall size of the correctional workforce or further investing in carceral infrastructure will require correctional officials to adopt additional changes that drastically shrink the total number of people held in various form of solitary confinement. Examples might include significantly revising disciplinary policies to permit the use of solitary confinement only as a last resort, in response to only the most serious disciplinary infractions, and for the shortest amount of time possible(4). Those persons previously housed for longer terms in solitary confinement could be diverted to more units repurposed to provide specialized treatment, particularly mental healthcare. As correctional agencies face the challenge of implementing legislation and other types of legal reforms, it is essential for state actors to recognize that incarcerated individuals who are deemed to require (or desire) physical separation from others due to serious health and safety reasons, need an environment led by compassionate professionals who are skilled at fostering meaningful social relationships, promoting engagement in clinical care, and creating professional interactions that are empathetic and supportive of people with complex mental health conditions shaped by violence, trauma, and socioeconomic adversities. 

Undoing “cultures of harm” and creating cultures of caring will also require authorities to critically identify and rescind existing policies that counteract the ethos and objectives of reform, such as those that implicitly authorize frontline officers to dehumanize incarcerated persons through deprivations of essential social and material resources, reliance on static security measures (e.g., restraints) that often result in psychological and bodily harms. Academic researchers working within the walls of correctional systems may turn to established theories of dehumanization and “moral disengagement” used to assess the structural forces, institutional policies, and interpersonal factors that shape state actors involvement in perpetuating human rights violations to create frameworks for critiquing exiting policies, understanding dynamics between officers and incarcerated persons, the range of harms solitary confinement inflicts, and creating organizational contexts to align solitary confinement reforms with international standards (4, 26, 50, 51).

In addition, reaching the full potential of the ORT model, in Oregon and other prison systems, may depend on the ability to achieve the goal of decarceration more generally—not only downsizing the prison system overall but also reducing the nation’s overreliance on carceral responses to crime and replacing it with more humane, social justice-oriented, public health-based approaches (52). The ORT’s transformation of a traditional solitary confinement unit from a culture of harm to a culture of caring conveys a broader message about what can be achieved when institutional systems are premised on recognizing the inherent dignity and humanity of the persons who are confined in them as well as those who work there. While shrinking prison populations mostly falls outside their legal powers, correctional staff can leverage their influence to call for system-level population reductions as means to foster safer and less stressful work environments that afford more opportunities for correctional staff to provide meaningful professional services and social support to meet the needs of people in their care and custody.”

e) Reviewer #2: Solitary confinement has been used as a form of judicially sanctioned punishment since the 1820s. Studies highlighting the negative health consequences associated with the use of isolation as a means of managing prisoners were first published in the 1840s. Despite continuing adverse reports from clinicians, psychologists, psychiatrists and non-governmental organisations highlighting the risks associated with limiting opportunities for prisoners to socialize with others, the solitary is widespread across many correctional systems. This is especially true of the US.

This submission reports on a recent attempt to implement a Norwegian Correctional Service initiative in the Oregon State Penitentiary, a maximum-security institution. This initiative was designed to improve rehabilitative outcomes for high-risk prisoners—especially those with serious mental illness (SRI). The program was limited to the Behavioral Health Unit (“BHU”) at the Oregon State Penitentiary. The inmates of the BHU have ‘serious and lengthy mental health histories and past instances of violence against themselves or others’. The case study describes and evaluates the work of the Oregon Resource Team (ORT) over a nine-month period in 2021.

We have revisited our description of the ORT intervention to make it clear that the program provided services to incarcerated persons within and outside the BHU. We also added language to Figure 1 that may be more informative and address Reviewer 2’s comment “g” below. 

f) The number of participants in the study is small. While 44 inmates took part in the ORT trial over the course of the study period, only 16 were interviewed. Administrative data could be accessed for all 44. These were supplemented with interviews with 14 correctional staff. The study uses this data to plot quarterly unit-level changes in staff uses of force from January 1, 2016 to December 31, 2021, a time period that includes the period before the implementation of the ORT experiment and its first 12 months of operation.

We agree that the number of participants in the feasibility stage of this pilot program may be considered relatively small by some, although it was sufficiently sized for an exploratory evaluation of an intervention in its pilot stage. We opted to use the 2016 to 2021 window, in part to account for potential variations that may have been due to disruptions related to institutional changes during the COVID-19 pandemic, which altered daily operations across the entire prison. We consider our qualitative analysis robust and sufficient to reach thematic saturation and is a sample size that is in line with many other qualitative studies. For example, please refer to a recent systematic review on this topic.

o Hennink, M., & Kaiser, B. N. (2022). Sample sizes for saturation in qualitative research: A systematic review of empirical tests. Social Science & Medicine, 292, 114523.

g) Although the study purports to report on the impact of the ORT experiment on the ‘behavior, health, and well-being of incarcerated persons and correctional staff, and ways to optimize its effectiveness and expand its use’ its major quantitative findings relate to falling levels of violence. Qualitative evidence from interviews is used to assess other positive impacts on the well-being of inmates and (interestingly) staff.

It would be particularly useful if the window in which the ORT experiment was implemented was marked on Figure 1. 

We agree that the changes suggested by the Reviewer will improve Figure 1. In response, we have revised Figure 1 to include a chronology of important events occurring over the observation period, and expanded upon the explanation in the figure’s legend to provide clarity on the combination of factors that likely accounted for the observed reductions in use-of-force incidents.

h) Even so it is apparent that the peak in use of force was in the last quarter of 2018 and that considerable amount of the reduction occurred in 2019 and especially 2020. Perhaps Table 2 could be reorganised to show the change in the use of force between years (eg. 2016, 17, 18, 19, 20 and 21 – the last being the year of the study). 

In our description of the development and implementation of the program, we provide a timeline of events leading up to 2021, when the program was fully-staffed. We now clarify that the fact that reductions began to occur in 2019 aligns with the series of events leading up to this point. Specifically: 1) correctional staff were trained on Norwegian correctional practices, which included use-of-force reduction training; 2) a group of officers began conducting activities with incarcerated persons in the unit, before the department had approved allocation of full-time positions; and 3) the facility leadership was actively advancing a larger set of initiatives focused on changing the culture throughout the institution to be more humane and dignity-driven. Please see our response to the item above and revisions to Figure 1. For this formative case-study, our goal was to describe and contextualize the reforms that were adopted and establish the possibility of plausible impacts of these changes with the aim of developing methodology for more rigorous evaluations of future Resource Team interventions. We have added language to the discussion section addressing the limitations of our analysis in dissecting the influence of each of these events and made recommendations for future studies to do so. 

i) More needs to be said about selection into the BHU over this time longer time frame and its operation. More needs to be said about the selection of individuals into the study.

We agree that readers would benefit from more clarity on each of these points. In the study methods, we now provide a detailed description of the various types of special housing units at the Oregon State Penitentiary. The BHU was the original primary focus of the ORT intervention when it began. We now explain that this was largely because the BHU is a unit that houses people with the most severe, acute psychiatric needs, who often struggle to adapt or decompensate in other housing units. 

Our research team had zero influence or control over who was assigned to the BHU (or any other unit); and the officers working for the ORT also had very little discretion over housing placements. Our study was retrospective, and one of our aims was to describe the characteristics and institutional histories of the group of incarcerated persons who ended up working with the ORT. As reported in the results, many of the incarcerated persons who engaged in ORT services were residing in other units within the special housing wing of OSP. Some had been sent to BHU from other institutions after a traumatic event.

The relevant revisions reads as follows:

• “The program began on a very small scale by focusing on just two persons who had been chronically isolated in the BHU. Based on its initial, albeit limited success, the ORT worked in tandem with Amend and their Norwegian colleagues to create new policies and procedures to facilitate program expansion. Starting in 2019, a team of ORT officers identified additional incarcerated persons in the BHU as potential participants. Selection was based on several factors, such as whether the person had experienced frequent and/or prolonged solitary confinement, had engaged in interpersonal violence with peers and/or staff, had instances of self-injury or suicidality, and/or had chronically declined or refused to leave their cell for programming, showering, or recreation. For example, people who were housed in other solitary confinement units in the ODOC, such as the “Intensive Management Units” at Two Rivers Correctional Facility and Snake River Correctional Institution, and who had recently been in situations involving acute psychiatric decompensation, self-injury, and/or multiple assaults against staff or other incarcerated persons that often escalated into uses of force by officers (e.g., cell extractions, chemical spray), before being transferred to the BHU were among those considered for inclusion. Although the pandemic limited the team’s ability to recruit new participants and conduct additional activities, by 2021, five full-time officers were funded to work with mental health and medical staff on an interdisciplinary ORT.”

• “Study Setting and Participants: As noted, the ORT project was piloted in the special housing units at OSP, the Oregon prison system’s main penitentiary, with a primary focus on the Behavioral Health Unit (BHU). The BHU is one of several special housing units in operation at OSP, all of which are intended for persons who are deemed “unable to adjust satisfactorily to the general population because of a serious mental illness.” The BHU (49 cells) is designated for “intensive behavioral management and skills training unit for inmates with serious mental illness that have committed violent acts or disruptive behavior,” all of whom are considered “unable to adjust satisfactorily to the general population because of a serious mental illness.” Another special housing unit, the Mental Health Infirmary (49 beds) is an acute “crisis response unit that provides psychiatric care and a therapeutic environment for inmates [sic] that require intensive assessment, care, and stabilization.” A third such unit, the Intermediate Care Housing (45 beds) is a “step down” unit for people transitioning out of the infirmary or BHU after stabilizing. The Day-treatment Unit (40 cells) permits residents to have more unstructured out-of-cell time, access to a dayroom, and is most similar to conditions in the general population. Finally, the Disciplinary Segregation Unit (65 beds) is punitive housing for people charged and convicted of rule violations, OR. Admin. R. 291-048-0210). 

The BHU was the main focus of the ORT, although several ORT participants were drawn from other OSP special housing units. It was selected for this intervention because its residents are considered by OSP staff to be among the most challenging to manage in the Oregon prison system and likely to be retained in long-term isolation. In many instances, their SMI diagnoses had contributed to frequent acts of self-injury and violent behavior. Many residents had been retained in solitary confinement-type conditions for years and had ceased participating in activities or utilizing out-of-cell time (including refusing showers and yard time). A number of them had manifested the extreme negative effects of solitary confinement by engaging in clinically dysfunctional and disruptive behavior (e.g., smearing feces on walls, flooding their cells with toilet or sink water, starting fires). All 14 ORT staff and all 44 incarcerated people who worked with the ORT during the evaluation period (January 1 through September 30, 2021) were eligible for qualitative interviews.” 

j) It might also be an idea to compare the ORT experiment with the outcomes of its Norwegian counterpart. Were they similar? A great deal of the submission is taken up with analysis of the interviews. These are informative but lengthy.

We agree that this would be an interesting direction for a future study, although we believe it is beyond the scope of this initial case study. 

k) This is an important study which could lead to significant reforms in current penal management but it needs to make sure that the statistical findings are robust and then use the qualitative evidence to flesh these out.

We agree that future studies should adopt a rigorous quantitative approach to evaluating the impact of this intervention. We believe that qualitative and descriptive methods were best suited to our research aims for this first, formative, exploratory case-study. Especially when drawing on administrative data not generated or collected for empirical research purposes, we would posit that collecting qualitative data was more essential as a first step to developing more robust quantitative data collection instruments for future studies and for developing methods for study design and statistical analysis, with larger sample sizes, that focus on quasi-experimental analytical approaches. As a case-study, the triangulation approach that we took has set the stage for us to conduct such research in the future. 

Reviewer #3: 

l) This is a spectacular piece of scholarship that I am excited to see (hopefully) published. I have very little in terms of edits as it is already well-written and rigorously presented. Just a couple of very small details to consider.

Thank you to reviewer 3 for your kind words. We are glad that you found the manuscript to be valuable and worthy of publication. 

m) At several places the authors use numerals rather than spelling out numbers. I'm not sure of the rules/guidelines for this journal but standard APA practice is to spell out numerals (not %) from zero to 11 and then use numerals for the rest. So 3, would be three. Totally minor, and a bit nitpicky. I apologize.

Thank you for pointing out this issue, we have addressed it in the current version. 

n) The literature review is fairly comprehensive but seems to be missing a new book that came out last year: Surviving Solitary by Rudes and colleagues. I'd consider working that in since the main message is about harm. I did see the citation of several pieces by Reiter though, and that's the other biggie to cite (in my view). Kudos.

Thank you. We agree that “Surviving Solitary” is an important, recent contribution and have added it as a reference. 

Thank you for your work and your passion for it. Loved reading this piece.

---

## [Decision Letter · Decision Letter 1]

21 Jun 2023

The Resource Team: A Case Study of a Solitary Confinement Reform in Oregon

PONE-D-23-00201R1

Dear Dr. Cloud,

I am pleased to inform you that your manuscript has been judged scientifically suitable for publication and will be formally accepted for publication once it meets all outstanding technical requirements.

Thank you for choosing PLOS One.

Kind regards,

Dr Nasrul Ismail

Academic Editor

PLOS ONE

Additional Editor Comments (optional):

N/.A

Reviewers' comments:

Reviewer's Responses to Questions

**Comments to the Author**

1. If the authors have adequately addressed your comments raised in a previous round of review and you feel that this manuscript is now acceptable for publication, you may indicate that here to bypass the “Comments to the Author” section, enter your conflict of interest statement in the “Confidential to Editor” section, and submit your "Accept" recommendation.

Reviewer #1: All comments have been addressed

Reviewer #2: All comments have been addressed

2. Is the manuscript technically sound, and do the data support the conclusions?

Reviewer #1: Yes

Reviewer #2: Yes

3. Has the statistical analysis been performed appropriately and rigorously? 

Reviewer #1: Yes

Reviewer #2: Yes

4. Have the authors made all data underlying the findings in their manuscript fully available?

Reviewer #1: No

Reviewer #2: No

5. Is the manuscript presented in an intelligible fashion and written in standard English?

Reviewer #1: Yes

Reviewer #2: Yes

6. Review Comments to the Author

Reviewer #1: Thank you for the opportunity to review this manuscript after revision. The authors have responded to all of my suggestions, and I think the article will make an important contribution to research on solitary confinement, corrections, health, and public policy.

Note that the authors cannot provide the underlying data for several important reasons. They provide sufficient reasoning to be allowed not to share their data publicly, and point people to the ways they can request data from the Oregon state prison system.

Reviewer #2: I have ticked No to 4 but the authors have provided good reasons why some of the underlying data cannot be made available. This is an important study and I am happy to recommend its publication following the addition of these revisions.

7. PLOS authors have the option to publish the peer review history of their article (what does this mean?). If published, this will include your full peer review and any attached files.

Reviewer #1: No

Reviewer #2: **Yes: **Hamish Maxwell-Stewart

---

## [Editor Report · Acceptance letter]

3 Jul 2023

PONE-D-23-00201R1 

The Resource Team: A case study of a solitary confinement reform in Oregon 

Dear Dr. Cloud:

I'm pleased to inform you that your manuscript has been deemed suitable for publication in PLOS ONE. Congratulations! Your manuscript is now with our production department. 

Kind regards, 

on behalf of

Dr. Nasrul Ismail 

Academic Editor

PLOS ONE